# Flavored e-cigarettes modulate embryo development, fetal growth, and potentiate early fetal demise without nicotine
Margeaux W. Marbrey [1] ✉, Samuel M. Cripps[1], Rennica Huang[1], Bryan M. Kistner[2], Aanvi Somany[1], Elizabeth S. Douglas [2] & Kathleen M. Caron [2]

## Abstract

**Background** Electronic cigarettes (e-cigarettes) function by aerosolizing a base liquid containing nicotine and flavoring, used by an estimated 15% of pregnant women as a supposed safer alternative to traditional cigarettes. Our previous studies demonstrated e-cigarettes can delay gestation. Limited studies have examined in vivo effects on the placenta.
**Methods** We exposed adult pregnant C57BL/6J female mice to flavored e-cigarettes with and without nicotine (VAPE NIC & VAPE). We measured implantation success ($N = 10$ SHAM, $N = 17$ VAPE, $N = 13$ VAPE NIC), erythrocyte presence ($N = 29$ SHAM, $N = 29$ VAPE, $N = 26$ VAPE NIC) and embryo elongation ($N = 25$ SHAM, $N = 29$ VAPE, $N = 22$ VAPE NIC) per implant site at day 6.5 at 13-21 weeks of age. Fetal and placental weight ($N = 11$ SHAM, $N = 14$ VAPE, $N = 12$ VAPE NIC) was evaluated at day 12.5 in mice aged 15–39 weeks, while placental gene expression was separately analyzed by offspring sex ($N = 7$ total, $N = 3$ sex-specific).
**Results** Here we show that e-cigarettes cause similar embryo elongation and in the absence of nicotine, exhibit elevated implant site blood cell accumulation which may contribute to fetal demise. With nicotine, e-cigarettes elicit a reduction in embryo to placental weight ratios. Genes involved in hypoxia, reactive oxygen species response, and placental growth including hypoxia inducible factor 1, alpha subunit (*Hif1a*), prostaglandin-endoperoxide synthase 2 (*Ptgs2*), glutathione peroxidase family members 2 and 3 (*Gpx2/Gpx3*), thioredoxin reductase 1 (*Txnrd1*), and mitogen-activated protein kinase 1 (*Mapk1*) exhibit marked decreases in placental tissue depending on fetal sex and nicotine presence.
**Conclusions** Our findings conclude flavored e-cigarettes modulate in vivo implantation and placentation mechanisms depending on the presence of nicotine. This work presents a measure of concern for flavored e-cigarette use during pregnancy.

## Plain language summary

Maternal smoking results in numerous adverse pregnancy outcomes. To avoid these dangers, women have started using electronic cigarettes or e-cigarettes as a perceived safer alternative. However, the safety of e-cigarette use during pregnancy is unknown. E-cigarettes heat an inhaled liquid containing glycerol, flavoring, and nicotine. Recent findings have determined e-cigarettes can impact early pregnancy. However, these studies are limited. Here we exposed mice to e-cigarette vapors containing flavoring with and without nicotine. We determined nicotine can impair fetal and placental growth, while the absence of nicotine may cause early miscarriage. Finally, we determined that several placental genes were changed depending on the presence of nicotine and fetal sex. Thus, we conclude e-cigarettes cause negative effects on pregnancy and should not be used.

Electronic nicotinic delivery systems (ENDS), also known as e-cigarettes, have escalated in popularity in 4.5% of adults and 11% of 18–24 year old individuals within the United States population[1]. Additionally, upwards of 15% of pregnant women in the US and UK have started using these devices as a supposed safer alternative to traditional cigarettes[2]. ENDS heat and aerosolize a base liquid of propylene glycol and vegetable glycerine containing varying additives such as nicotine, flavorings, and thickening agents[3]. Lifestyle factors such as traditional cigarette smoking impart serious effects on pregnancy outcomes and *in utero* exposed offspring, including fetal growth restriction, behavioral abnormalities, implantation failure and miscarriage[4,5]. Interestingly, smoking cigarettes is correlated with protection from hypertensive pregnancy disorders such as preeclampsia due to an unknown mechanism[6]. Yet, our understanding of how ENDS impact early pregnancy and placenta development is lacking, especially in the context of

[1]Duke University School of Medicine, Department of Obstetrics & Gynecology, Division of Reproductive Sciences, 701W Main Street, Suite 510, Durham, NC, 27701, USA. [2]University of North Carolina Chapel Hill School of Medicine, Department of Cell Biology & Physiology, 111 Mason Farm Road, Chapel Hill, NC, 27599, USA. ✉e-mail: Margeaux.marbrey@duke.edu

flavorings, a preference of pregnant women[7]. Early implantation is characterized by a molecularly primed uterus that receives the developed embryo at day 4.5 in mice and day 9.5 in women[8]. Late delivery of the fertilized embryo to the uterine epithelium can result in fetal demise or inability to implant[9,10]. Furthermore, post-implantation embryo growth failure can often lead to miscarriage or adverse pregnancy outcomes[8]. As implantation continues, blood cells develop only in the yolk sac starting at day 8 in mouse development[11]. Thus, the presence of blood cells or erythrocytes in the implantation capsule is often indicative of early fetal death or dysregulation[8]. After implantation, the fetal trophoblast cells migrate through the uterine tissue and elicit recruitment of immune cells and vascular remodeling enzymes, to renovate the maternal vasculature into high-capacity vessels promoting maximal exchange of gases and nutrients[12]. Specialized immune cells known as uterine natural killer cells marked with dolichos biflorus agglutinin or DBA-LECTIN staining[13], are involved in the remodeling of the maternal spiral arteries into high-capacity vessels[14]. These vascularization changes can be visualized by endothelial cell marker, platelet and endothelial cell adhesion molecule 1 or PECAM1, expressed readily in endothelium from healthy placental tissues[15]. Due to this rapid growth, the placenta experiences dynamic fluctuations in proliferation, reactive oxygen species, and hypoxia. Dysregulation of these processes is known to correlate with early pregnancy loss, a staggering outcome of 60% of pregnancies[16]. Further, impaired implantation or placental malformation can result in hypertensive pregnancy disorders and pre-term birth with elevated risk for maternal and fetal deaths[17]. Thus, understanding the factors that regulate early pregnancy mechanisms is paramount in developing targets to treat disease and improve patient care. As lifestyle factors such as tobacco exhibit known roles in modulating pregnancy mechanisms, it is important to understand how this new ENDS device impacts early pregnancy success. Our previous findings determined ENDS exposure permits fertility yet delays gestational timing[18]. In a murine model fertility trial, pups were born 3–4 days late. When natural timed matings were recorded, no implantation sites were identified at day 5.5, the late stage of implantation. Transcriptomic analyses suggested multiple receptivity pathways were changed (NCBI GEO: GSE131077). However, we were not able to assess how ENDS impacted mid-gestation from day 6.5 to day 21. Since the placenta is sensitive to a variety of external stimuli, we determined to examine how ENDS might affect mid-gestation.

Currently, few studies have examined the effects of ENDS on placental development. One study determined immortalized trophoblast cells administered ENDS liquids exhibited impaired invasion and tubal formation[19], while another recent study demonstrated impaired cell viability and placental factors in human placental explant tissue treated with ENDS liquid components[20]. To address these limitations in the field, we utilized natural, timed matings and whole-body exposure chambers to examine the effects of ENDS on post-implantation and placental development. We determined to extend our studies from our previous work to examine days 6.5 and 12.5 of pregnancy, representing post-implantation and placental development timepoints. Therefore, this study describes how flavored ENDS modulate late implantation and placenta formation. Using a wildtype mouse model, we conclude that flavored ENDS exposure elicits specific defects in growth restriction, gene expression, and implant site blood cell accumulation that may potentiate future fetal demise.

## Methods

### Rodent model of e-cigarette exposure

All animal experiments were performed and ethically approved under the animal care protocol (#21-083) by the Institutional Animal Care and Use Committee at the University of North Carolina, Chapel Hill. Animals were provided two types of enrichment, including nestlets and Tecniplast Mezzanines, with food and water provided *ad libitum* and housed in a 12-hour light cycle. No unexpected or adverse events occurred, and minimization of pain, suffering, and distress was prioritized. No humane endpoints were utilized. Wildtype C57BL/6J females (in-house maintained colony originally sourced from The Jackson Laboratory Strain #000664) were exposed to sham (SHAM) or e-cigarette (ENDS) vapors containing flavoring with 6 mg/mL of nicotine (VAPE NIC) and without nicotine (VAPE) for 3 hours per day, 5 days a week, for at least three weeks to simulate ENDS use independent of pregnancy. The SciReq inExpose System was employed using an ENDS mod device, the Joyetech Mini at 65 Watts at 250 °C. ENDS liquids employed were strawberry custard flavored and commercially available: VapeTasia Killer Kustard Strawberry (0 mg/mL nicotine=VAPE group, 6 mg/mL nicotine=VAPE NIC group). Although these liquids exhibit a lower amount of nicotine compared to previous publications (24 mg/mL)[18], the liquids are commercially used by individuals due to the popular flavor. The authors specifically incorporated e-cigarette liquids that might be used by pregnant individuals. Furthermore, the presence of a matched non-nicotine liquid allows for comparison of the effects of nicotine versus flavoring. After three weeks of ENDS exposure, females were mated to wildtype males. Upon the presence of the copulatory plug (day 0.5 of pregnancy), timed, naturally mated animals were exposed daily during pregnancy until day 6.5 or day 12.5. For the day 6.5 cohort, whole reproductive tracts were fixed flat in 4% paraformaldehyde overnight, washed in phosphate-buffered saline (PBS), and processed and embedded in paraffin blocks. Day 6.5 dams were euthanized between 13–21 weeks of age, depending on exposure start time and time of plug. The average age at the time of euthanasia for the SHAM and VAPE groups was 18 weeks, and for VAPE NIC, 17 weeks. Littermates were distributed across treatment groups for comparable experimentation. For the day 12.5 cohort, fetal and placental tissues were dissected and weighed. Dams at day 12.5 with two or fewer viable fetal-placental pairs were excluded from analysis. Day 12.5 dams were euthanized between 15-39 weeks of age, depending on exposure start time and time of plug. Average age at the time of euthanasia for SHAM: 29 weeks, VAPE: 28 weeks, and VAPE NIC: 31 weeks. Placentas were sliced in half and either flash frozen for RNA isolation or processed and embedded in paraffin blocks. All samples were separately and distinctly measured. No samples were measured repeatedly.

### Measuring implant site elongation and erythrocyte presence

Whole paraffin blocks were sectioned at 5 µM and every slide was collected and chronologically numbered. Every fifth slide was stained for morphology using hematoxylin and eosin traditional staining methods. Imaging was completed at 5x magnification on a Zeiss Axio Imager.A2 microscope. Implant sites were categorized and scored and the slide with the longest implant elongation region per embryo was imaged and measured using Fiji. Implanted embryos were each measured from the anti-mesometrial to mesometrial polar head. Measurements were analyzed for each treatment group and plotted. Embryo implant sites were measured from $N = 3$ dams per treatment group. The difference from the mean was determined by subtracting data points from the mean of the respective group. One-way ANOVA with Tukey's multiple comparisons test was implemented to determine significance. Grubb's outlier test was used to exclude outlier data points when appropriate. The coefficient of variation was determined by dividing the standard deviation by the average and multiplying by 100 to generate a percentage of variation for each treatment group. Embryo implant sites were scored for the presence of blood cells or erythrocytes. Negative presence of erythrocytes was scored as none. Positive presence of erythrocytes was scored as either low accumulation or high accumulation, specifically in the implant capsule.

### Serum cotinine ELISA

Blood samples were obtained by retro-orbital bleed at necropsy and serum isolated using BD Microtainer Blood Collection Tubes (Becton Dickinson #3655967) by centrifuging for 10 minutes at 3000 rpm and stored at −80 °C until further use. The Cotinine Direct ELISA kit (Aviva Systems Biology #GWB-BQK0DA) was used to measure cotinine levels in serum according to the manufacturer's instructions (SHAM: $N = 9$; VAPE: $N = 12$; VAPE NIC: $N = 10$). Each sample was assayed in duplicate. The sensitivity of the kit reported by the manufacturer was 1 ng/mL. All samples that had no detectable cotinine and had the same absorbance as the kit standard for 0 ng/

mL of cotinine were assigned the same value for statistical analysis. GraphPad Prism was used to fit a standard curve from the kit standards to interpolate cotinine concentrations (R squared = 0.98). One VAPE NIC sample was excluded as the absorbance values indicated that cotinine concentrations were too high to be interpolated by the model. One-way ANOVA with Tukey's multiple comparisons test was implemented to determine significance using GraphPad Prism. Results expressed as mean ± standard error of the mean (SEM).

### RNA isolation
Placental halves isolated from different dams were homogenized using an MP Biomedicals FastPrep Bead homogenizer with Invitrogen™ TRIzol™ Reagent (ThermoFisher Cat # 15596026). Phase separation was performed twice, first with 1-Bromo-3-chloropropane (Sigma #B9673-200ML) and then with chloroform (Fisher Sci #C6074). RNA was precipitated using 100% ethanol and purified using a column from the Qiagen RNeasy® Mini Kit (Qiagen Cat #74104). RNA was *DNase1* treated with 1 mg/mL DNAse (Roche #10104159001) for 10 minutes and re-purified on the Qiagen RNeasy® Mini Kit. Reverse Transcription was performed using High-Capacity cDNA Reverse Transcription Kit with RNase Inhibitor (ThermoFisher Cat. #4374966). Quantitative Real Time PCR was performed on the Applied Biosystems™ QuantStudio™ 6 Flex Real-Time PCR System (ThermoFisher Cat #4485692) using Applied Biosystems™ PowerUp™ SYBR™ Green Master Mix for qPCR (ThermoFisher Cat#A25776) with validated SYBR primers determined from Harvard Primer Bank according to manufacturer's instructions. SYBR primers utilized are listed in Supplemental Table 1.

### Fetal sex identification
Fetal sex was determined by removing a small biopsy of the day 12.5 fetus and purifying the DNA using a solution of 100 mM Tris (pH 8.0), 5 mM EDTA, 0.2% SDS, 200 mM NaCl, and 0.167 mg/mL proteinase K (Macherey-Nagel #740506). SRY genotyping was used to determine sex, using the primers: SX_F, 5′-GATGATTTGAGTGGAAATGTGAGGTA-3′, SX_R, 5′-CTTATGTTTATAGGCATGCACCATGTA-3′ according to published methods[21]. EconoTaq DNA polymerase was utilized for PCR with manufacturer's instructions (Lucigen #30031-1).

### Immunofluorescence and quantification of placental morphological features
Staining of fixed and paraffin-embedded placental halves was performed according to previously published methods[22] by fixing cut half placentas with 4% paraformaldehyde overnight before storage in phosphate-buffered saline (PBS) at 4 °C. Tissues were processed and embedded in paraffin wax blocks and sectioned at 5 µm. Tissues were dehydrated, permeabilized, and antigen retrieval was performed with 10 µM sodium citrate with 0.05% Tween. Tissue was blocked in 10% normal goat serum diluted in phosphate-buffered saline solution with 0.01% Tween (PBS/T). Primary antibody for platelet and endothelial cell adhesion molecule 1 (PECAM1) (Millipore Sigma MAB1398Z) was used at 1:150 concentration. Secondary antibodies for dolichos biflorus agglutinin (DBA-LECTIN, VWR #101097-972, 1:100 dilution), Cy3-goat, anti-Armenian hamster (VWR 102646-984, 1:200 concentration), and Hoechst (Thomas Scientific #33258, 1:250 dilution) were implemented in above mentioned PBS/T. Tissue was mounted with Prolong Gold Antifade (Life Technologies #P36934) before sealing with clear nail polish. Negative controls with no primary and no secondary antibodies were implemented. Images were acquired on the Leica Thunder Imager DMi8 microscope. Fiji was used to measure the area of the placental layers according to the demarcations described in Supplemental Fig. 1a. Uterine natural killer cells were quantified by counting the respective DBA-LECTIN positive cells in a demarcated box within each respective placental decidua. One-way ANOVA with Tukey's multiple comparisons test was implemented to determine significance.

### Statistics and reproducibility
Replicates for Figs. 1 and 2 were defined as litter numbers or weights, dams, individual embryo implantation sites, resorption numbers, or individual fetal-placental pairs. Individual placentas were depicted as replicates for gene expression for Fig. 3 or for area quantification in Supplementary Fig. 1. Fiji image analysis was used to quantify embryo elongation and placental layer size. Leica Aivia software was used to count uterine natural killer cell number. GraphPad Prism and Microsoft Excel were used to tabulate data, perform statistical tests, and generate graphs. Implant site number, embryo length, embryo and placental weights, embryo/placental weight ratios, and resorption numbers were compared using one-way ANOVA with Tukey's multiple comparisons test to determine significance between groups. Resorption incidence was determined by scoring the number of resorptions per litter and categorizing the litter as exhibiting 0, 1, 2, or ≥ 3 resorptions. The number of categorized uteri per treatment group was divided by the total uteri characterized as 0,1,2, or ≥ 3 resorptions to generate a percentage. The percent of litters exhibiting a specific number of resorptions per treatment group was calculated as parts of whole. Embryo difference from the mean was measured using one-way ANOVA with Tukey's multiple comparisons test. Fisher's exact test was used to measure the significance of the resorption incidence. Gene expression was measured using two-way ANOVA with Tukey's multiple comparisons test. The GraphPad Outlier test was utilized when appropriate. Sample sizes were determined according to Power Analysis and relying on our experience in the field measuring pregnancy parameters. Blinding of animal exposures was employed during necropsies and analyses to eliminate bias. Mouse identifying numbers were generic and did not correspond to treatment conditions. Confounding variables were limited by exposing animals at the same time in the same methods across exposure groups. Littermates were randomly chosen and distributed evenly across treatment groups to control for age and litter differences. No computer randomization was employed.

### Reporting summary
Further information on research design is available in the Nature Portfolio Reporting Summary linked to this article.

## Results
### E-cigarette exposure promotes serum cotinine levels with no change in litter size
Wildtype C57BL/6J females were primed with sham (SHAM) or e-cigarette (ENDS) vapors containing flavoring with 6 mg/mL nicotine (VAPE NIC) or without nicotine (VAPE) for 3 hours per day, 5 days per week. After three weeks, females were timed and naturally mated to C57BL/6J proven males, and at the presence of the copulatory plug (day 0.5), females were exposed daily during pregnancy until necropsy on day 6.5. The litter size was tabulated for each dam and no differences were identified between exposure groups (Fig. 1a). Whole blood was obtained at necropsy, and serum cotinine levels were measured across groups. Serum concentrations of cotinine from mice exposed to nicotine (VAPE NIC) ($124.7 ± 7.04$ ng/mL) were significantly higher compared to that of mice exposed to base liquid with 0 mg nicotine (VAPE) ($1.35 ± 0.91$ ng/mL; $p = 0.008 × 10^{-12}$) and of the SHAM mice ($0.24 ± 0.24$ ng/mL; $p = 0.008 × 10^{-12}$) (Fig. 1b). Cotinine serum concentrations were not significantly different between SHAM and VAPE groups ($p = 0.9746$) (Table 1).

### Exposure to e-cigarettes results in implant site erythrocyte accumulation and similar elongation of embryos
Whole uteri were fixed and sectioned and every fifth slide was stained for morphology using hematoxylin and eosin (Fig. 1c–e). Embryo implant sites exposed to ENDS without nicotine exhibited increased erythrocyte accumulation compared to SHAM controls (Fig. 1f). Furthermore, implant sites exposed to nicotine demonstrated fewer erythrocytes compared to SHAM controls. Finally, after morphological staining, embryos were individually identified in their longest elongation state as a proxy of embryo development

**Table 1 | Cotinine serum concentrations from e-cigarette exposed mice**

| Treatment group (N) | Cotinine serum levels (mean ± SEM ng/mL) | P-value |
|---|---|---|
| VAPE NIC (6 mg nicotine) (N = 9) | 124.7 ± 7.04 | $p = 0.008 \times 10^{-12}$ [a,b] |
| VAPE (0 mg nicotine) (N = 12) | 1.35 ± 0.91 | $p = 0.9746$ [c] |
| SHAM (N = 9) | 0.24 ± 0.24 | |

Statistical testing was performed by one-way ANOVA with Tukey's multiple comparisons test with data expressed as the mean ± standard error of the mean (SEM) ng/mL. $p$ = adjusted $p$-value. (SHAM, VAPE NIC N = 9, VAPE N = 12).
[a] VAPE NIC vs. VAPE.
[b] VAPE NIC vs. SHAM.
[c] VAPE vs. SHAM.

and measured and graphed (examples depicted in Fig. 1g, h). Elongation or development of embryos in the SHAM control was variable from 64 μm to ~580 μm in length (Fig. 1i). However, exposure to ENDS without nicotine caused a narrowing of elongation from ~118–500 μm. Interestingly, exposure to nicotine tightened this window even further, with many of the embryos concentrated to the mean of ~248 μm. To assess the variability between these groups, we calculated the coefficient of variation and determined the SHAM group exhibited a 50.92% variation compared to the VAPE group with a 38.11% variation and the VAPE NIC group with a 31.34% variation. To further account for this narrowing effect, the difference from the respective group mean was calculated for each data point and graphed (Fig. 1j). The ENDS groups demonstrated significant changes compared to SHAM using one-way ANOVA with Tukey's multiple comparisons test (SHAM vs VAPE $p = 0.0164$; SHAM vs VAPE NIC $p = 0.00257 \times 10^{-1}$; VAPE vs VAPE NIC $p = 0.2831$). Thus, exposure to ENDS can elicit accumulation of erythrocytes and promote homogeneity in embryo elongation.

**Exposure to e-cigarette vapors reduces embryo-to-placental weight ratios across fetal-placental units**

Embryos and placentas were dissected and weighed from pregnant animals exposed until day 12.5 with ENDS vapors. Embryo and placental weights and weight ratios were either averaged per litter and graphed (Fig. 2a–c) or graphed individually as a fetal-placental unit (Fig. 2d). Averaged embryo and placental weights per litter did not exhibit any differences across ENDS treatment groups compared to SHAM (Fig. 2a, b). Averaged embryo and placental weight ratios per litter did not exhibit significant differences, albeit lowered in the VAPE NIC exposed group (SHAM vs VAPE NIC $p = 0.0968$) (Fig. 2c). However, fetal-placental pairs individually graphed embryo to placental weight ratios were significantly reduced in only the VAPE NIC group (SHAM vs VAPE NIC $p = 0.0365$; VAPE vs VAPE NIC $p = 0.0231$) (Fig. 2d). Additionally, the presence of resorptions was calculated at day 12.5. The number of fetal resorptions between groups exhibited no difference (Fig. 2e). The percentage of fetal absorption incidence per uteri across groups appeared to be elevated in the VAPE group without nicotine (Fig. 2f, g). Further, in the presence of nicotine, VAPE NIC, less resorptions seem to have occurred. However, these differences were not significant using Fisher's exact test ($p = 0.1784$). Although weights were not different between groups, placental layers were evaluated for changes in decidua, labyrinth, or junctional zone area. To measure the area of the placental layers, whole placentas were stained for dolichos biflorus agglutinin or DBA-LECTIN, PECAM1, and Hoechst. Area of layers was measured using Fiji demarcated in Supplemental Fig. 1a with representative placental images depicted in Supplemental Fig. 1b–d. No differences were found in placental layer sizes across groups (Supplemental Fig. 1e). Additionally, DBA-LECTIN positively stained uterine natural killer cells were counted in a grid in the maternal decidua and no difference was observed across groups (Supplemental Fig. 1f).

**Placental development genes exhibit sex and exposure-specific downregulation**

Placenta tissues at day 12.5 were grouped by sex. RNA was isolated and gene expression was examined across all treatment groups in the context of fetal sex. Hypoxia inducible factor 1, alpha subunit (*Hif1a*) was downregulated in compiled placentas in only the VAPE group (SHAM vs VAPE $p = 0.0358$) (Fig. 3a). Furthermore, females were specifically susceptible to downregulation of prostaglandin-endoperoxide synthase 2 (*Ptgs2*) (compiled: SHAM vs VAPE $p = 0.0244$; SHAM vs VAPE NIC $p = 0.0319$; female: SHAM vs VAPE $p = 0.0071$; SHAM vs VAPE NIC $p = 0.0327$) (Fig. 3b). Peroxidase genes in the glutathione peroxidase family, glutathione peroxidase 2 (*Gpx2*) and 3 (*Gpx3*), were decreased in compiled and female placentas (*Gpx2* compiled: SHAM vs VAPE NIC $p = 0.0224$; female: SHAM vs VAPE NIC $p = 0.0288$; *Gpx3* compiled: SHAM vs VAPE $p = 0.0438$; SHAM vs VAPE NIC $p = 0.0480$) (Fig. 3c-d). Oxidative stress responder, thioredoxin reductase 1 (*Txnrd1*) was also significantly decreased in the VAPE group for compiled and male placentas (compiled: SHAM vs VAPE $p = 0.0046$; male: SHAM vs VAPE $p = 0.0372$) (Fig. 3e). Mitogen-activated protein kinase 1 (*Mapk1*) was decreased depending on sex and exposure group (compiled: SHAM vs VAPE $p = 0.0238$; SHAM vs VAPE NIC $p = 0.0011$; male: SHAM vs VAPE $p = 0.0201$; female: SHAM vs VAPE NIC $p = 0.0493$) (Fig. 3f). SYBR primers utilized are listed in Supplementary Table 1.

**Discussion**

The use of ENDS is prevalent among pregnant women, often thought of as a safe substitute to traditional cigarettes[2]. Despite this perceived safety, few studies have examined how ENDS with flavoring affect an in vivo pregnancy. These studies utilize a mouse model of exposure to assess the safety of flavored ENDS on post implantation and placentation. We determined flavored ENDS without nicotine caused accumulation of erythrocytes within implant site regions, with both exposure groups exhibiting similar lengths of embryo elongation. In addition, flavored ENDS with nicotine caused decreases in fetal/placental weight ratios when comparing the individual fetal-placental pairs. Both ENDS exposure groups elicited varying downregulation of critical placental development genes depending on fetal sex. Finally, ENDS without nicotine may impact resorption rate, while nicotine presence may protect embryos from early fetal loss; yet further studies are needed to confirm these effects on resorption. We concluded that ENDS cause marked accumulation of erythrocytes in the implant sites, which may be an indicator of early pregnancy loss as the embryo capsule is often devoid of blood cells[11]. Furthermore, embryo elongation exhibits homogeneity in ENDS-exposed groups, especially in the presence of nicotine. This suggests there may be a delay in oocyte or embryo development before implantation or a delay in oviductal transport to the uterus[23]. The elongation measurements of the ENDS with the nicotine group suggest the embryos may be contained in the oviduct longer, yet this requires further investigation. These findings correlate with our previous findings suggesting ENDS may be causing an arrival time delay, as the implanted embryos were not identified at day 5.5, the time of implantation[18]. Thus, further investigation is required to examine how ENDS modulate early implantation events. Impaired placental development can contribute to fetal growth restriction[24]. In corollary, women who use cigarettes during pregnancy often experience lower birth weights[25]. However, nicotine replacement therapy has been shown to induce higher birth weights[5]. We identified that ENDS with nicotine result in decreased fetal/placental weight ratios when comparing the individual fetal-placental unit, thus suggesting even minor levels of nicotine combined with ENDS can have fetal weight consequences. We also measured resorption number at mid-gestation and identified that ENDS without nicotine may cause an increased incidence of resorptions compared to the other groups; yet additional studies and a larger sample size are required to achieve a statistically significant result. This trend is not unique to these studies, as nicotine has often resulted in protective outcomes in vascular-mediated mechanisms[5]. Although traditional cigarette smoking is protective of hypertensive pregnancy disorder risk, nicotine is not

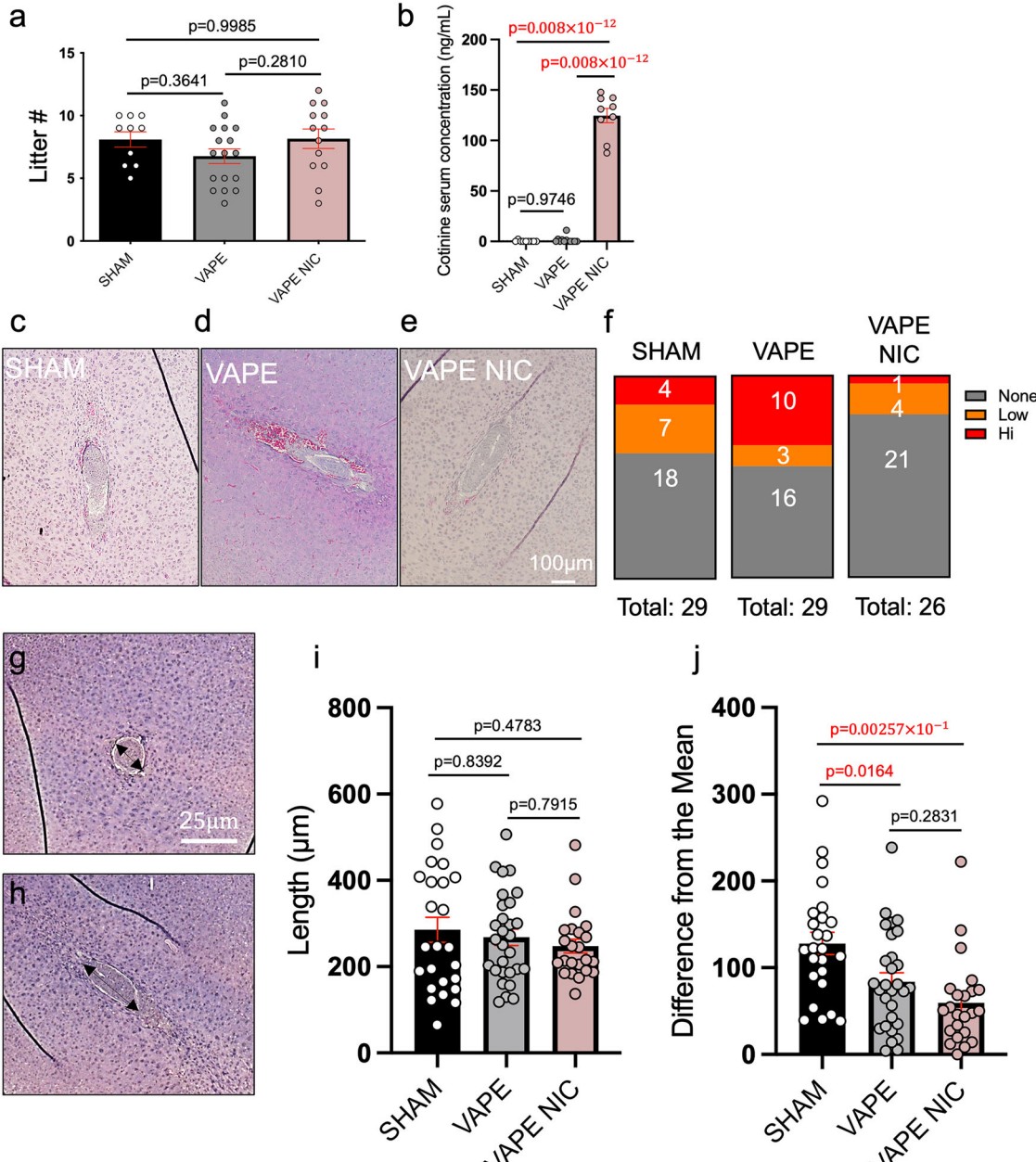

**Fig. 1 | E-cigarette exposure causes accumulation of erythrocytes and similar embryo elongation at post-implantation.** Implantation sites were scored at day 6.5 after exposure to SHAM or e-cigarette vapors with nicotine (VAPE NIC) and without nicotine (VAPE). Total litter number per dam was counted per exposure group and analyzed with one-way ANOVA with Tukey's multiple comparisons test (**a**) ($N = 10$ SHAM, $N = 17$ VAPE, $N = 13$ VAPE NIC). Serum cotinine levels were measured across groups and analyzed via one-way ANOVA with Tukey's multiple comparisons test (**b**) ($N = 9$ SHAM, $N = 12$ VAPE, $N = 9$ VAPE NIC). Longitudinal hematoxylin & eosin-stained representative implant site regions are depicted (**c-e**) (scale bar=0.1 mm) and erythrocyte presence was blindly scored per embryo with red bars indicative of high erythrocyte numbers, orange with low erythrocyte numbers, and grey with no erythrocyte presence (**f**) ($N = 3$ dams per group, implant site number $N = 29$ SHAM, $N = 29$ VAPE, $N = 26$ VAPE NIC). Elongation measurements of developing embryos were obtained by staining for hematoxylin and eosin and measuring the length of the embryo capsule using Fiji. Example embryo measurement images are depicted with double-sided arrows indicating the measured length of the embryo capsule (**g, h**). Length of elongating embryos was determined (**i**) ($N = 3$ dams per group, implant site number $N = 25$ SHAM, $N = 29$ VAPE, $N = 23$ VAPE NIC) and difference from the mean was measured with one-way ANOVA with Tukey's multiple comparisons test (**j**) ($N = 3$ dams per group, implant site number $N = 25$ SHAM, $N = 29$ VAPE, $N = 22$ VAPE NIC). Grubbs outlier test was used where appropriate. Error bars indicate mean ± SEM. Red font p-values indicate significance with a $p$-value ≤ 0.05. SHAM depicted as white circles on black bars (**a, b, i, j**); e-cigarette vapors without nicotine (VAPE) depicted as grey circles on grey bars (**a, b, i, j**); e-cigarette vapors with nicotine (VAPE NIC) depicted as pink circles on pink bars (**a, b, i, j**). p=adjusted $p$-value.

considered the major player as these protective mechanisms are not present with smokeless tobacco products[5]. These data suggest ENDS with nicotine cause mild growth restriction, yet also simultaneously protect the fetus from early loss. Further studies are required to understand how nicotine causes this protective mechanism at mid-gestation. In addition to nicotine, metabolites in the ENDS flavoring may also elicit dysregulation of pathways important for embryo implantation and placental development. Flavoring compounds have been reported to cause cytotoxic effects, inflammation, and increased levels of reactive oxygen species[26]. As early pregnancy initiation is highly sensitive to levels of reactive oxygen species and cellular toxins, the presence of e-cigarette vapor-associated toxicants can interrupt processes of pregnancy that could result in adverse pregnancy outcomes.

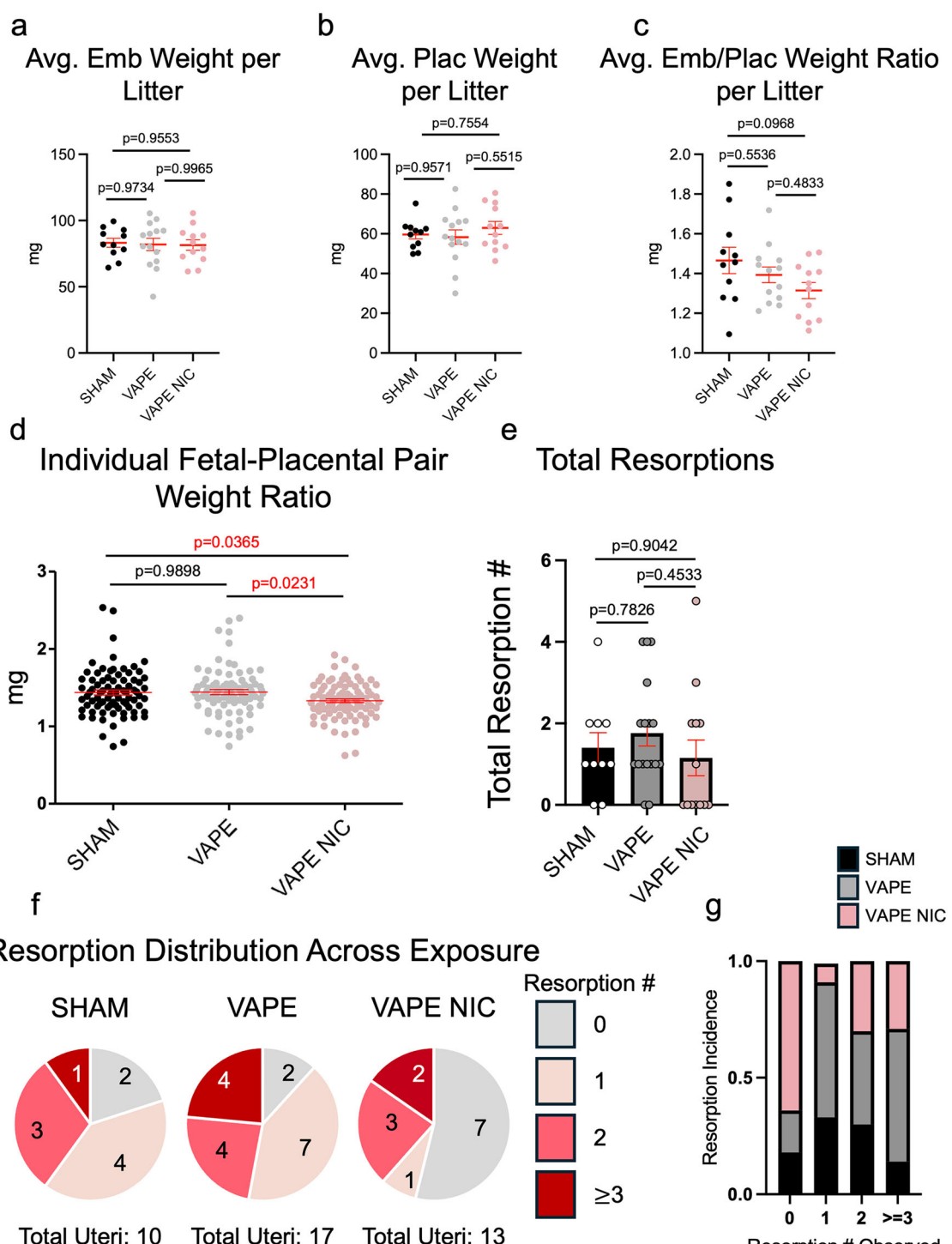

**Fig. 2 | Exposure to e-cigarette vapors decreases embryo-to-placental weight ratios across the fetal-placental unit and may promote resorptions in the absence of nicotine.** After exposure to e-cigarette vapors until day 12.5 of pregnancy, embryo and placental weights were measured and averaged per litter and analyzed using one-way ANOVA with Tukey's multiple comparisons test (**a, b**) ($N = 11$ SHAM, $N = 14$ VAPE, $N = 12$ VAPE NIC). Embryo/placental weight ratios were averaged per litter and analyzed using one-way ANOVA with Tukey's multiple comparisons test (**c**) ($N = 11$ SHAM, $N = 13$ VAPE, $N = 12$ VAPE NIC). Embryo/ placental weight ratios were also depicted individually as fetal-placental pairs and analyzed with one-way ANOVA and Tukey's multiple comparisons test with a global ANOVA $p$-value $= 0.0125$ (**d**) ($N = 82$ SHAM, $N = 86$ VAPE, $N = 90$ VAPE NIC). Total resorptions or fetal death observed per uterus was graphed and tested

using one-way ANOVA with Tukey's multiple comparisons test (**e**) ($N = 10$ SHAM, $N = 17$ VAPE, $N = 13$ VAPE NIC). Resorption incidence across exposure was depicted as parts of a whole (**f–g**) (total uteri scored, $N = 10$ SHAM, $N = 17$ VAPE, $N = 13$ VAPE NIC) with (**f**) grey representing 0 resorptions, light pink as 1 resorption, pink as 2 resorptions, or dark red as $\geq 3$ resorptions. SHAM depicted as black circles (**a–d**), white circles on black bars (**e**), or black bars (**g**); e-cigarette vapors without nicotine (VAPE) depicted as grey circles (**a–d**), grey circles on grey bars (**e**), or grey bars (**g**); e-cigarette vapors with nicotine (VAPE NIC) depicted as pink circles (**a–d**), pink circles on pink bars (**e**), or pink bars (**g**). Error bars indicate mean ± SEM. Red font $p$-values indicate significance with a $p$-value $\leq 0.05$. $p$ adjusted $p$-value, Emb embryo, Plac placenta.

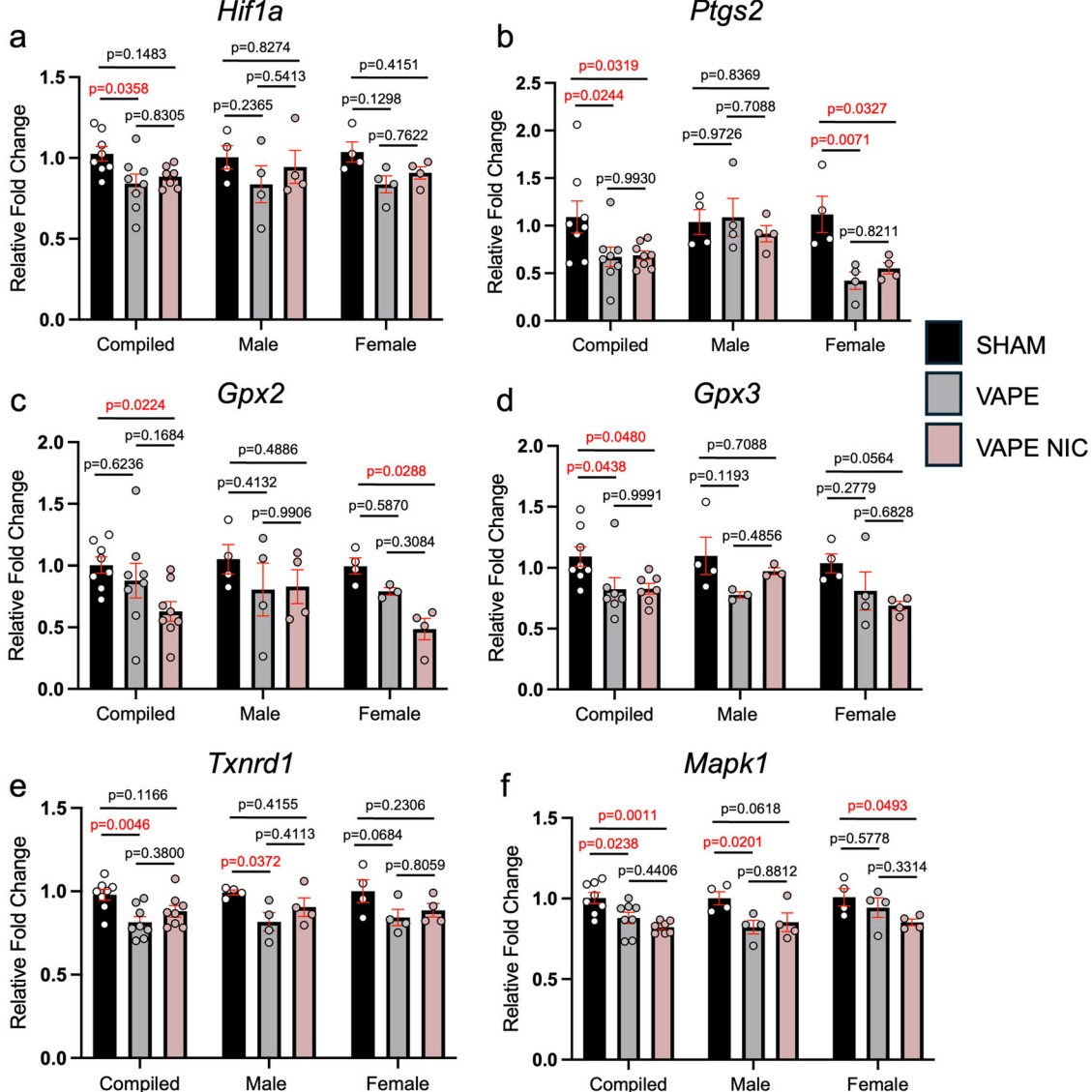

**Fig. 3 | E-cigarettes elicit differential gene expression dependent on fetal sex.** Gene expression was performed on e-cigarette-exposed, sexed placental tissue at day 12.5 (**a–f**). Two-way ANOVA was performed with Tukey's multiple comparisons test. Grubbs' Outlier test was used when appropriate. SHAM depicted as white circles on black bars; e-cigarette vapors without nicotine (VAPE) depicted as grey circles on grey bars; e-cigarette vapors with nicotine (VAPE NIC) depicted as pink circles on pink bars (**a–f**). Error bars indicate mean ± SEM. $p$ = adjusted $p$-value. Red font $p$-values indicate significance with a $p$-value ≤ 0.05. $N$ numbers: *Hif1a* (compiled: $N = 8$ SHAM, $N = 8$ VAPE, $N = 7$ VAPE NIC; male, female: $N = 4$), *Ptgs2* (compiled: $N = 8$; male, female: $N = 4$), *Gpx2* (compiled: $N = 8$; male: $N = 4$; female: $N = 4$ SHAM, $N = 3$ VAPE, $N = 4$ VAPE NIC), *Gpx3* (compiled: $N = 8$ SHAM, $N = 7$ VAPE, $N = 7$ VAPE NIC; male: $N = 4$ SHAM, $N = 3$ VAPE, $N = 3$ VAPE NIC; female: $N = 4$), *Txnrd1* (compiled: $N = 8$; male, female: $N = 4$), *Mapk1* (compiled: $N = 8$ SHAM, $N = 8$ VAPE, $N = 7$ VAPE NIC; male, female: $N = 4$).

Furthermore, exposure to ENDS may also impact pre-fertilization mechanisms, including oocyte development, lordosis, and secreted hormone circulations. Although this work was unable to examine these mechanisms, future studies are required to identify how ENDS mechanistically impact early pregnancy. Despite finding no morphological differences in the sizes of the placental layers, we identified numerous changes in placental gene expression. We examined genes involved in proliferation, hypoxic response, and reactive oxygen species response as these represent important pathways in placental development[17]. We determined gene expression changes varied greatly depending on fetal sex. These data suggest fetal sex can contribute to gene expression changes in the presence of ENDS *in utero* exposure. Our previous studies identified specifically female offspring exposed *in utero* to ENDS with nicotine caused chronic metabolic deficits in adulthood[18]. Since these studies were not examined to term, we were not able to conclude how ENDS with flavoring impact *in utero*-exposed adults. Current published studies suggest flavorings cause long-

term implications in exposed offspring[27]. Further studies are warranted to understand how specifically ENDS flavorings can impact offspring.

Animal models have often been used to model *in utero* nicotine exposure but can present challenges. For instance, it's estimated that serum cotinine levels in pregnant women who smoke traditional cigarettes range from 99–246 ng/mL[28]. In rodent models of nicotine exposure, doses have centered around 1–2 mg/kg per day, which was determined to be consistent with pregnant smokers consuming 20 cigarettes each day[29]. However, recent ultra-performance liquid chromatography-tandem mass spectrometry in female wildtype mice exposed to ENDS for 4 weeks, 1 hour per day with a 25 mg/mL nicotine concentration, determined cotinine levels of $362.3 \pm 16.27$ ng/mL[30]. This suggests e-cigarette exposures may exhibit higher levels of nicotine in rodent models compared to traditional cigarettes. It is also known that pregnant animals clear nicotine faster than non-pregnant animals[31]. In a recent paper using male rodents, mice were treated with flavored ENDS and

exhibited cotinine levels of 150.44 ± 22 ng/mL when treated with 36 mg/mL of nicotine[32]. We exposed our animals for 3 hours per day instead of 1 hour. Thus, although our e-liquid exhibited less nicotine, the animals exhibited a longer exposure, and we reported a mean nicotine level of 124.7 ± 7.04 ng/mL. Although the animals in this study exhibited a higher serum cotinine level compared to the non-nicotine containing group or SHAM group, they exhibited a lower range of nicotine exposure. This lower nicotine level is a limitation of this study, especially as the nicotine levels of currently used e-cigarette disposables on the market are 50 mg/mL. Additionally, the mouse as a model for these exposure studies exhibits inherent limitations. As measured in rodent adult and fetal brains[33,34], the mouse requires a 10-fold nicotine concentration to match the effects observed in the rat. These differences are likely due to variable respiration rates between the rat and mouse. Additionally, outbred and inbred mouse strains can exhibit differential nicotine tolerance, which can further confound results and comparability among existing studies[34]. Thus, nicotine exposure studies in the mouse exhibit limitations that should be noted. Therefore, these reported findings represent a low level of nicotine exposure in an inbred mouse strain. This suggests that today's ENDS with higher nicotine concentration are more potent and may exhibit more severe health outcomes; yet, further studies are needed to confirm this. This work presents an initial understanding of how flavored ENDS impact post-implantation development and placental formation. We concluded that ENDS with flavoring elicit accumulation of erythrocytes and homogeneity of embryo development which could be a precursor to early pregnancy loss or a delay in embryonic development. Additionally, fetal to placental unit weight ratios were smaller in the ENDS nicotine group, yet the absence of nicotine may induce embryonic resorption. Further, the presence of nicotine may protect against resorption incidence; yet additional confirmatory studies are required. Finally, ENDS elicited decreases in placental gene expression that varied depending on fetal sex. Therefore, flavored ENDS modulate the health and growth of embryos and placentas which may exhibit implications for growth restriction or early pregnancy loss. Thus, the use of ENDS devices should be cautioned during pregnancy.

## Data availability

All data that support the findings of this study are available within the paper and its Supplementary Information. Figure 1a, b, f, i, j source data is described in Supplementary Data 1, Fig. 2a–g source data is found in Supplementary Data 2, and Fig. 3a–f source data is described in Supplementary Data 3. Each figure with corresponding source data is listed in the specified Supplementary Data file in a demarcated figure tab. Supplementary Fig. 1e, f source data is reported in Supplementary Data 4. Primers used for gene expression studies are listed in Supplementary Table 1. All other data are available from the corresponding author upon reasonable request.

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

## Acknowledgements
The authors acknowledge support from the laboratories of Kathleen Caron, Margeaux Marbrey, Claire Doerschuk, and Susan Murphy and from Carolyn Suitt and the CGIBD Histology Core at the University of North Carolina. This work was supported by NIH/NICHD grants: R01HD060860 (K.M.C.), 1K99HD10490001 (M.W.M.), R00HD104900 (M.W.M.).

## Author contributions
M.W.M. performed most of the experiments, conceptualization, management, and writing of the manuscript. K.M.C. provided guidance on conceptualization, execution, management, and final manuscript edits. S.M.C., R.H., B.M.K., and A.S. performed experiments and contributed to final figures. E.S.D. provided laboratory support, experiment execution, and guidance.

## Competing interests
The authors declare no competing interests.
