## [Transparent Peer Review file · Communications Medicine]

Flavored E-Cigarettes Modulate Embryo Development, Fetal Growth, and Potentiate Early Fetal Demise Without Nicotine

Corresponding Author: Dr Margeaux Marbrey

Version 0:

Reviewer comments:

Reviewer #1

(Remarks to the Author)

The manuscript of Marbrey et al. is focused on the study of the effects of electronic nicotinic delivery systems (ENDS) on placenta and the embryonic implantation process in mice. There are scarce studies on this issue available. This study can have a high impact on public health, since an increased percentage of pregnant women are using these devices assuming they have a safer health outcome compared to the classic cigarette smoking. In global terms, the manuscript describes clearly the question to be addressed and the approach to answer the question. However, there are several caveats as described below.

Major concerns:

1) Why the authors used nicotine concentration of 6 mg/ml to be administered with ENDS? In other studies exploring in vitro and ex vivo the effects of nicotine and ENDS on placenta and implantation, nicotine concentrations were 24 mg/mL (Wetendorf et al., 2019, doi: 10.1210/js.2019-00216) and 12 mg/ml (Sáez-Villanueva et al., 2018, doi: 10.1016/j.reprotox.2018.07.084).

2) In mice, the dosing regimen is 10-fold higher than that used in rats to achieve similar upregulation of [3H]-nicotine binding in the fetus, which is considered the gold standard for nicotine actions when administered during intrauterine life. By contrast, in rats nicotine 6 mg/ml is OK to upregulate nicotine receptors. These higher doses required in mice, are related to differences in nicotine metabolism: the plasma half-life for nicotine is 5–7 min in mice, whereas 54 min in rats and 2–2.5 h in humans (doi: 10.1111/j.1440-1681.2009.05214.x.). This point is crucial, because a mouse model could be a wrong election to perform this study, leading to negative results only because nicotine dose is low.

The authors should provide the plasma steady state concentration of nicotine reached with their regimen in dams. When nicotine is administered through osmotic minipumps, the plasma nicotine concentration able to upregulate the nicotine receptors, without affecting the size of litters and the growth curve of mouse neonates, is approximately 240 ng/mL.

Comparison with steady state concentrations attained by other studies using ENDS should be discussed.

3) Statistical analysis

In Figure 1 data treatment is confused. In Fig 1A, for example, it seems that were originated from 3 dams per group?, but the plotted variable is the number of implanted embryos per horn. May be for that reason statistical analysis was not performed. The authors should increase the number of experiments to increase the N=3 per group (using the average value per dam as N=1 for each condition). Recommend avoid the use of this mixture of paired and unpaired data, which is not appropriated. This kind of problem is observed along several of the graphics shown in Figs 1 and 2.

In Fig 1E, Chi-square analysis may be useful.

In Fig.1 H and 1 I, similar statistical problems arise. There is a mixture of paired and unpaired data. Which advantage offers Fig 1 I? It is clear from standard deviations that there are differences in the dispersion of data among the three groups. As indicated in line 112-113, the use of one-way ANOVA with the unpaired student t-test is not appropriated because within each group there are data coming only from 3 dams. And the Mann-Whitney test requires all the data to be unpaired.

In Figure 2. It is not clear the N used in each plot. In the legend appears number of dams per group, but the number of data plotted overpass the number of dams, which means that several dots in each group come from a same dam. Please clarify this inconsistency. Average data from each dam should be used as N=1.

I recommend in the legend of figures incorporate the information of real N and the statistical test used in each situation to get a global difference and the post hoc test used for multiple comparisons. Please indicate associated to each graph the p

value obtained with the global ANOVA.

4) Several explanations and inferences are not totally or uniquely supported by the results themselves. In several parts of the manuscript the authors say that “exposure to ENDS can promote synchrony in embryo elongation”. Why the authors use the word “synchrony” instead of homogeneity? The narrowing of the range of embryonic lengths can be explained by time-independent factors.

The authors say in discussion, line 203: “This work defines a foundational understanding of how flavored ENDS impacts post implantation development and placental formation”. However, this work reports the possible effects of ENDS on some features of placenta and implantation processes. That is, this asseveration should be based on mechanisms on which a defined component of ENDS is affecting placental formation and implantation.

Discussion should be revised avoiding expressions not supported by results. Additionally, discussion of possible mechanisms through which VAPE without and with nicotine may generate their impacts on the placenta and embryo implantation should be provided.

Minor points

- 1) In lines 121-123 the sentence: “The number of fetal resorptions trended higher in the VAPE group compared to SHAM and the VAPE NIC groups” should be omitted since no statistical difference support it.
- 2) In line 99, the authors report no change in litter number and reference Fig 1A. However, Fig 1A was built on basis of number of implanted embryos per horn, which is misleading. Please reserve the litter number to indicate “the number of young animals born to an animal at one time”.
- 3) In methodolo

Reviewer #2

(Remarks to the Author)

Overall, this study is based on a really important area of research as while smoking is clearly harmful to pregnancy outcomes, much less is known in relation to Vaping. It is also really important to look at vaping constituents as nicotine along has controversially been shown have some positive effects as well as negative effects. In contrast, next to nothing is known about other inhaled vape components and the authors have used what appears to be a suitable method exposing mice to the vapour from these products. While this type of study is highly valuable, the methodological approaches to investigate embryo and placental outcomes do have their limitations which are discussed in the comments below.

1. The study is largely descriptive with gene expression and basic placental morphology assessed. Why were the gene changes not followed up by protein analysis. It would have also been interesting to look at a range of other factors involved in placental development, fetal growth, hormone synthesis. Perhaps, even an unbiased RNAseq screen would have been highly beneficial in a study such as this. Given the changes that were identified at the RNA level, it would have been vital to look at things like tissue oxygenation using products like pimonidazole, or tissue oxidative stress markers. Did these changes in gene expression impact placental vascularisation which could have been assessed histologically (more than just labyrinth size).

2. The summary of what you have shown previously in your 2019 paper should be better explained in the introduction to further support your current paper and to make it clear what is new and better about the current study vs what was done before- The previous paper nicely explains what was undertaken and this should be summarised in more detail in the current paper. Explain how your fertility studies involved placing the mouse in with the male for 4 months while exposed to vape and that the 3 to 4 days delay was to littering (maybe it took longer to mate, rather than longer gestation) and the lack of implantation sites at E5 may also be explained by reduced mating success or mismatch in ovarian cycle and uterine cycle). This is important as your current study uses an approach where the male is removed after the presence of a plug.

3. Following on from previous comment- How many days were the females left in with the male before mating in the current study? While you have not explicitly, it is clear that these are not time mated overnight so gestational age is entirely dependant on the accuracy of plug indicating day 0.5 of pregnancy. This is highly unusual for studies looking at tissues at time points as early as E6.5 as it is quite important that the gestational age is tightly controlled.

4. Is it possible that the pre pregnancy vape exposure has impacted estrous cycling, oocyte quality, or mating behaviour? In the previous work, the male was also exposed- could this have impacted outcomes?

4. How have you statistically represented your litters in this study. You seem to have measured indivial embryo lengths, individual fetus weights and placental weight and yet the entire litter was exposed to the vape and so that statistical unit should be the litter. You have completed statistical analysis for several of these parameters and talk about “trends” when you have n=28 embryos but only from 3 litters. This is not powered sufficiently and you really only have n=3. You also need to add details about number of litters exposed per group, how many samples per litter were analysed and how the individual pups were included in the analysis (ie one per litter or using nested statistical analysis)

5. How were the placentas selected for RNA analysis? Was this one male and one female per litter?

Reviewer #3

(Remarks to the Author)

This is a well written manuscript describing the effects of ENDS vapor with flavor (+/- nicotine) on implantation, embryo development and placental development. A few minor comments below:

*In the Abstract, the statement should be changed from "genes important to placental development were changed" to "expression of genes important for placental development were changed..."

*The Introduction needs more information to understand the implications of the results. For example, in the paragraph on placental development, add information about the implications of erythrocyte accumulation at the implantation site. Add information on the importance of DBA, LECTIN and PECAM1. Add information about what is known about embryo elongation and offspring outcomes.

*More information is needed in the methods section to describe how erythrocyte levels were categorized as high and low.

Version 1:

Reviewer comments:

Reviewer #1

(Remarks to the Author)

The authors have improved the manuscript, in particular the discussion, but major concerns persist:

1. Use of mice to be exposed to ENDS with nicotine concentration of 6 mg/ml.

The authors incorporate in the new manuscript the study of cotinine serum concentration, which indicates that only dams exposed to vape plus nicotine exhibit this nicotine metabolite in blood. They also inform that the levels of cotinine in vape plus nicotine-exposed dams were in the lower range, and, likely, the serum concentrations of nicotine were also in the low range. This new result makes more relevant to discuss the limitations of using mice instead of rats to be exposed to nicotine 6 mg/ml, since this dose is ten times inferior to that required in mice for getting similar effects as those in rats. As indicated previously, this point is crucial, because a mouse model could be a wrong election to perform this study, leading to negative results only because nicotine dose is low. At least, this must be extensively discussed.

2. Statistical analysis

In methods, the description of statistical analysis is not clear. In Figures and legends, what authors considered N value for statistics is confusing. They show a mixture of data showing an individual fetal-placental unit as N = 1 (Fig 2D), and in other graphs, they use, as recommended by statistics, the average value per dam as N=1 for each variable (for example, Fig. 2C).

1) There are conclusions that are not supported by statistical analysis of data:

In Figure 2 and in the Results and Discussion, author say that "exposure to E-cigarette vapors decreases embryo to placental weight and promotes resorptions in the absence of nicotine". The average embryo/placenta weight ratio in Fig 2C exhibits no difference ($P > 0.05$), while the analysis performed in Fig 2D is not valid since it uses an individual fetal-placental unit as N = 1. In Fig. 2E the number of total resorptions are not different among the three experimental groups, and Figs. 2F and 2G do not have statistics.

2) In Fig. 1J, the variable illustrated is the "difference from mean". Why the authors use this untraditional parameter, that doesn't take count of differences in the mean, instead of coefficient of variation, which is the (standard deviation/mean) ?

3) I recommend a Two-Way ANOVA analysis for data in Figure 3 (male- female versus sham-vape-vape nic).

Minor points (number of lines as in revised version in word).

1) In line 518 replace "synchronized" with "similar".

2) In line 134: what do you mean with "coordinated"? Do you want to say "similar length of embryo elongation"?

Reviewer #2

(Remarks to the Author)

The authors have addressed my major concerns and clarified some key concerns such as each litter representing n=1. For many parameters, the litter was appropriately used as the N. For others, the embryo was the N but now with information making it clear that it was from N=3 dams, the reader can interpret the data how they see fit. Thank you for providing additional information about the model.

Reviewer #3

(Remarks to the Author)

All points from the initial review have been successfully addressed.

Version 2:

Reviewer comments:

Reviewer #1

(Remarks to the Author)

In this new version, the authors have addressed satisfactorily all the risen points

Response to Reviewers' comments:

Reviewer #1 (Remarks to the Author):

The manuscript of Marbrey et al. is focused on the study of the effects of electronic nicotinic delivery systems (ENDS) on placenta and the embryonic implantation process in mice. There are scarce studies on this issue available. This study can have a high impact on public health, since an increased percentage of pregnant women are using these devices assuming they have a safer health outcome compared to the classic cigarette smoking. In global terms, the manuscript describes clearly the question to be addressed and the approach to answer the question. However, there are several caveats as described below.

Major concerns:

1) Why the authors used nicotine concentration of 6 mg/ml to be administered with ENDS? In other studies exploring in vitro and ex vivo the effects of nicotine and ENDS on placenta and implantation, nicotine concentrations were 24 mg/mL (Wetendorf et al., 2019, doi: 10.1210/js.2019-00216) and 12 mg/ml (Sáez-Villanueva et al., 2018, doi: 10.1016/j.reprotox.2018.07.084).

The reviewers are correct that this nicotine level is lower than previous publications. We desired to incorporate a commercially available and popularly used e-cigarette liquid containing flavoring and nicotine that might be used among pregnant individuals. Additionally, this commercial e-cigarette liquid was chosen as there was a flavor-matched no-nicotine (0mg) liquid that could be used as a control for these studies. The manuscript has been updated with this rationale (Lines 354-358) and description of this limitation (lines 311-319).

2) In mice, the dosing regimen is 10-fold higher than that used in rats to achieve similar upregulation of [3H]-nicotine binding in the fetus, which is considered the gold standard for nicotine actions when administered during intrauterine life. By contrast, in rats nicotine 6 mg/ml is OK to upregulate nicotine receptors. These higher doses required in mice, are related to differences in nicotine metabolism: the plasma half-life for nicotine is 5–7 min in mice, whereas 54 min in rats and 2–2.5 h in humans (doi: 10.1111/j.1440-1681.2009.05214.x.). This point is crucial, because a mouse model could be a wrong election to perform this study, leading to negative results only because nicotine dose is low.

The authors should provide the plasma steady state concentration of nicotine reached with their regimen in dams. When nicotine is administered through osmotic minipumps, the plasma nicotine concentration able to upregulate the nicotine receptors, without affecting the size of litters and the growth curve of mouse neonates, is approximately 240 ng/mL. Comparison with steady state concentrations attained by other studies using ENDS should be discussed.

We thank the reviewer for the comment. Other studies in mice have used e-liquids containing 0-36 mg/mL with a male cotinine level of 150.4 +- 22 ng/mL and have stated

in their publications that it represented “the highest concentration available on the US market” (Noel 2020, PMID 32083945). Smoking serum cotinine levels in pregnant women range between 99-246 ng/mL (Hickson et al. 2018). Yet, due to the constantly changing e-cigarette market, 50 mg/mL is now the standard concentration commercially available and often most seen in disposables, the most popular e-cigarette today.

Our purpose in this study was to implement an e-cigarette liquid that was popular in use and on the lower end of nicotine to attract the pregnant consumer. With the passing of time and flux of the e-cigarette market, these studies represent a low nicotine exposure, although our animals exhibit significantly upregulated serum cotinine at 124.7 ± 7.04 ng/mL (Figure 1B, Table 1). Although other animal models would be more advantageous for these studies, our laboratory specifically employs the mouse to serve as a benchmark for our transgenic mouse mechanistic studies. We thank the reviewer for their insightful comments. The discussion has been updated to include this discourse (lines 296-319).

3) Statistical analysis

In Figure 1 data treatment is confused. In Fig 1A, for example, it seems that were originated from 3 dams per group?, but the plotted variable is the number of implanted embryos per horn. May be for that reason statistical analysis was not performed. The authors should increase the number of experiments to increase the N=3 per group (using the average value per dam as N=1 for each condition). Recommend avoid the use of this mixture of paired and unpaired data, which is not appropriated. This kind of problem is observed along several of the graphics shown in Figs 1 and 2.

In Fig 1E, Chi-square analysis may be useful.

In Fig.1 H and 1 I, similar statistical problems arise. There is a mixture of paired and unpaired data. Which advantage offers Fig 1 I? It is clear from standard deviations that there are differences in the dispersion of data among the three groups. As indicated in line 112-113, the use of one-way ANOVA with the unpaired student t-test is not appropriated because within each group there are data coming only from 3 dams. And the Mann-Whitney test requires all the data to be unpaired.

There is no paired data in this manuscript. All animals were treated and necropsied. All measurements were taken at necropsy or from processed, harvested tissues. The embryo elongation data are specifically plotted per embryo as much variation exists between embryos within the same horn and litter due to natural variation within implantation developmental timelines (*Mantalenakis and Ketchel, 1966, PMID 5951130*). To average the embryos together per dam for embryo elongation would result in a wide standard error and broad spread of data due to normal differences between embryos. Thus, the elongation and erythrocyte data are intrinsic to the individual embryo capsule. The current figure description accurately depicts individual embryos per group resulting in more refined and informative data.

In Figure 2. It is not clear the N used in each plot. In the legend appears number of dams per group, but the number of data plotted overpass the number of dams, which

means that several dots in each group come from a same dam. Please clarify this inconsistency. Average data from each dam should be used as N=1.

Figure 2 A-C has been re-analyzed per dam while Figure D represents the individual fetal-placental pairs. Figure descriptions have been updated for clarity.

I recommend in the legend of figures incorporate the information of real N and the statistical test used in each situation to get a global difference and the post hoc test used for multiple comparisons. Please indicate associated to each graph the p value obtained with the global ANOVA.

Global anova p values have been reported in each figure legend when appropriate.

4) Several explanations and inferences are not totally or uniquely supported by the results themselves. In several parts of the manuscript the authors say that “exposure to ENDS can promote synchrony in embryo elongation”. Why the authors use the word “synchrony” instead of homogeneity? The narrowing of the range of embryonic lengths can be explained by time-independent factors.

The text has been updated in response to the reviewer’s comments (Lines 41, 143, 165, 235, 322).

The authors say in discussion, line 203: “This work defines a foundational understanding of how flavored ENDS impacts post implantation development and placental formation”. However, this work reports the possible effects of ENDS on some features of placenta and implantation processes. That is, this asseveration should be based on mechanisms on which a defined component of ENDS is affecting placental formation and implantation.

Discussion should be revised avoiding expressions not supported by results. Additionally, discussion of possible mechanisms through which VAPE without and with nicotine may generate their impacts on the placenta and embryo implantation should be provided.

The text has been edited to address the reviewer’s comments. Additional discussion has been added to address these topics (lines 270-280).

Minor points

1) In lines 121-123 the sentence: “The number of fetal resorptions trended higher in the VAPE group compared to SHAM and the VAPE NIC groups” should be omitted since no statistical difference support it.

The text has been edited to remove this statement (lines 184).

2) In line 99, the authors report no change in litter number and reference Fig 1A. However, Fig 1A was built on basis of number of implanted embryos per horn, which is

misleading. Please reserve the litter number to indicate “the number of young animals born to an animal at one time”.

3) In methodolo

The text has been reformatted for clarity (line 132) and Figure 1A has been changed to litter number to limit confusion.

Reviewer #2 (Remarks to the Author):

Overall, this study is based on a really important area of research as while smoking is clearly harmful to pregnancy outcomes, much less is known in relation to Vaping. It is also really important to look at vaping constituents as nicotine along has controversially been shown have some positive effects as well as negative effects. In contrast, next to nothing is known about other inhaled vape components and the authors have used what appears to be a suitable method exposing mice to the vapour from these products. While this type of study is highly valuable, the methodological approaches to investigate embryo and placental outcomes do have their limitations which are discussed in the comments below.

1. The study is largely descriptive with gene expression and basic placental morphology assessed. Why were the gene changes not followed up by protein analysis. It would have also been interesting to look at a range of other factors involved in placental development, fetal growth, hormone synthesis. Perhaps, even an unbiased RNAseq screen would have been highly beneficial in a study such as this. Given the changes that were identified at the RNA level, it would have been vital to look at things like tissue oxygenation using products like pimonidazole, or tissue oxidative stress markers. Did these changes in gene expression impact placental vascularisation which could have been assessed histologically (more than just labyrinth size).

We did not observe visible differences in placental vascularization using PECAM1 staining depicted in Supplemental Figure 1. Placental hemodynamics and oxidation were not measured in real time in these animals. Since the necropsies have been completed and these studies are no longer ongoing, it is outside the scope of this manuscript. We agree with the reviewer that future studies including placental hemodynamics, oxidative stress, and RNA sequencing would be informative and would further identify differentially regulated molecular signaling pathways in the context of vape exposure.

2. The summary of what you have shown previously in your 2019 paper should be better explained in the introduction to further support your current paper and to make it clear what is new and better about the current study vs what was done before- The previous paper nicely explains what was undertaken and this should be summarised in more detail in the current paper. Explain how your fertility studies involved placing the mouse in with the male for 4 months while exposed to vape and that the 3 to 4 days delay was to littering (maybe it took longer to mate, rather than longer gestation) and the lack of implantation sites at E5 may also be explained by reduced mating success or

mismatch in ovarian cycle and uterine cycle). This is important as your current study uses an approach where the male is removed after the presence of a plug.

We thank the reviewer for their kind comments. Additional background information describing the previous paper has been incorporated into the introduction (lines 90-96). It should be noted that the day 5.5 studies in the previous paper and the day 6.5 studies in this current manuscript were performed similarly as natural timed mating experiments. The difference is the day of gestation and the type of e-cigarette aerosol (previous: 24 mg/mL nicotine, 55:45 ratio of propylene glycol:vegetable glycerin versus current manuscript: 6 mg/mL nicotine, 30:70 ratio of propylene glycol: vegetable glycerin + added unknown proprietary flavoring compounds).

3. Following on from previous comment- How many days were the females left in with the male before mating in the current study? While you have not explicitly, it is clear that these are not time mated overnight so gestational age is entirely dependant on the accuracy of plug indicating day 0.5 of pregnancy. This is highly unusual for studies looking at tissues at time points as early as E6.5 as it is quite important that the gestational age is tightly controlled.

The day 12.5 and day 6.5 studies were timed natural matings with proven males and daily early morning copulation plug checks to determine mating time, defined as day 0.5. Females were left with males for a series of 3-4 days with daily morning plug checks to ensure accurate gestational time. Once a plug was observed, females were promptly removed from breeding and recorded. Experimental details are described in the introduction, results, and methods for further clarity (lines 91-96, 101-105, 130, 360).

4. Is it possible that the pre pregnancy vape exposure has impacted estrous cycling, oocyte quality, or mating behaviour? In the previous work, the male was also exposed- could this have impacted outcomes?

For this manuscript, no male stud counterparts were exposed to vape. We chose to limit these studies to only female reproductive outcomes. Although mating behavior, estrous cycling, and oocyte quality were not measured, ENDS vapors may impact these mechanisms. Although outside the scope of this work, we have mentioned this possibility in the discussion (lines 273-280).

4. How have you statistically represented your litters in this study. You seem to have measured individual embryo lengths, individual fetus weights and placental weight and yet the entire litter was exposed to the vape and so that statistical unit should be the litter. You have completed statistical analysis for several of these parameters and talk about "trends" when you have n=28 embryos but only from 3 litters. This is not powered sufficiently and you really only have n=3. You also need to add details about number of litters exposed per group, how many samples per litter were analysed and how the individual pups were included in the analysis (ie one per litter or using nested statistical analysis)

Figure 1A has been modified to report the litter number per dam, per group. All embryo elongation data and erythrocyte accumulation data are intrinsic to the embryo implant capsule. Due to the natural developmental variability between embryos within the same uterine horn, to average these data per dam would potentiate loss of these details. In Figure 2, fetal and placental weight data has been reanalyzed per dam (2A-C) and ratios are shown both per dam and across fetal-placental unit (2D). The n numbers and methods have been updated accordingly.

5. How were the placentas selected for RNA analysis? Was this one male and one female per litter?

Placental n number is listed in the figure legends for RNA analysis. The n number is also unique to each dam as female and male placentas were isolated and analyzed from different dams to ensure depth and confidence of results. The methods have been updated to reflect this (line 408).

Reviewer #3 (Remarks to the Author):

This is a well written manuscript describing the effects of ENDS vapor with flavor (+/- nicotine) on implantation, embryo development and placental development. A few minor comments below:

*In the Abstract, the statement should be changed from "genes important to placental development were changed" to "expression of genes important for placental development were changed..."

The abstract has been modified according to the reviewer's comments.

*The Introduction needs more information to understand the implications of the results. For example, in the paragraph on placental development, add information about the implications of erythrocyte accumulation at the implantation site. Add information on the importance of DBA, LECTIN and PECAM1. Add information about what is known about embryo elongation and offspring outcomes.

Additional information has been added to the introduction (lines 63-69, 73-77).

*More information is needed in the methods section to describe how erythrocyte levels were categorized as high and low.

Additional description is provided in the methods section regarding erythrocyte scoring (lines 380-384).

We thank the reviewers for their additional comments. We have implemented changes to the manuscript for increased clarity and transparency. We have additionally modified the figures and figure legends with clarified headings and appropriate statistical tests and updated p-values according to the reviewers' comments below. We have also updated the supplemental figure as we noticed the images were out of focus. Please see the below discussion regarding the reviewers' comments:

Reviewers' comments:

Reviewer #1 (Remarks to the Author):

The authors have improved the manuscript, in particular the discussion, but major concerns persist:

1. Use of mice to be exposed to ENDS with nicotine concentration of 6 mg/ml.

The authors incorporate in the new manuscript the study of cotinine serum concentration, which indicates that only dams exposed to vape plus nicotine exhibit this nicotine metabolite in blood. They also inform that the levels of cotinine in vape plus nicotine-exposed dams were in the lower range, and, likely, the serum concentrations of nicotine were also in the low range. This new result makes more relevant to discuss the limitations of using mice instead of rats to be exposed to nicotine 6 mg/ml, since this dose is ten times inferior to that required in mice for getting similar effects as those in rats. As indicated previously, this point is crucial, because a mouse model could be a wrong election to perform this study, leading to negative results only because nicotine dose is low. At least, this must be extensively discussed.

We appreciate the reviewer's comments and concerns. We have updated the discussion with a section describing this limitation (lines 303-311).

2. Statistical analysis

In methods, the description of statistical analysis is not clear. In Figures and legends, what authors considered N value for statistics is confusing. They show a mixture of data showing an individual fetal-placental unit as N = 1 (Fig 2D), and in other graphs, they use, as recommended by statistics, the average value per dam as N=1 for each variable (for example, Fig. 2C).

We have added further details in the methods and figure legends to transparently report the sample sizes accurately and limit confusion. We have also modified the figure titles for further clarity.

As mentioned previously for Figure 2, we think there is a benefit to observe the data as both fetal-placental pair and as individual litters representing the n sample size. To note, for the data with individual litters represented (Figure 2A-C), we have previously listed in the figure legend and methods that we are reporting data from greater than 11 litters per group: N=11 SHAM, N=14 VAPE, N=12 VAPE NIC. For the fetal-placental pair analysis (Figure 2D), we have previously

reported greater than 80 fetal-placental pairs represented per group: N=82 SHAM, N=86 VAPE, N=90 VAPE NIC.

1) There are conclusions that are not supported by statistical analysis of data:

In Figure 2 and in the Results and Discussion, author say that “exposure to E-cigarette vapors decreases embryo to placental weight and promotes resorptions in the absence of nicotine”. The average embryo/placenta weight ratio in Fig 2C exhibits no difference ($P > 0.05$), while the analysis performed in Fig 2D is not valid since it uses an individual fetal-placental unit as $N = 1$. In Fig. 2E the number of total resorptions are not different among the three experimental groups, and Figs. 2F and 2G do not have statistics.

We think there is value in observing and reporting the data as both per litter and by fetal-placental unit. With that said, Figure 2D does report a decrease in embryo to placental weight ratio. Figure 2F and 2G indicate the incidence of resorptions, not the number of resorptions. The incidence appears to be trending higher in the VAPE NIC group compared to the other groups. Since the sample size is too small for Chi-Square analysis, we have performed Fisher’s exact test which reports a non-significant result. We have updated the results and discussion to address the limitations of these data. We have edited the figure legend and manuscript to transparently report the data and highlight the significant and non-significant results (lines 147-166, 202-225, 241-249, 318-321, 628-640).

2) In Fig. 1J, the variable illustrated is the “difference from mean”. Why the authors use this untraditional parameter, that doesn’t take count of differences in the mean, instead of coefficient of variation, which is the (standard deviation/mean) ?

The authors noticed the embryos exposed to VAPE or VAPE NIC were clustered close to the mean. The SHAM group exhibited variability across the mean. The averages were similar across groups, but the VAPE exposed embryos were surprisingly similar and clustered at the mean. This could suggest the embryos are developing more similarly or may exhibit a transport delay which allows for this pause and timed development. These findings were described previously in the results (lines 131-138) and the discussion (lines: 226-236). Since the mean was central to our observations, we concluded the difference from the mean would be a relevant measurement. In addition to Figure 1J, per the reviewer’s suggestion, we have reported the percent variation, or the coefficient of variation listed in the results section (lines 138-141).

3) I recommend a Two-Way ANOVA analysis for data in Figure 3 (male- female versus sham-vape-vape nic).

We have updated Figure 3 with Two-Way ANOVA analysis for the female and male qRT-PCR comparisons. We have removed data on *Pgf* and *Nr2f2* as they no longer exhibited significant results.

Minor points (number of lines as in revised version in word).

1) In line 518 replace “synchronized” with “similar”.

2) In line 134: what do you mean with “coordinated”? Do you want to say “similar length of embryo elongation”?

The manuscript text has been revised to address these minor points.

Reviewer #2 (Remarks to the Author):

The authors have addressed my major concerns and clarified some key concerns such as each litter representing $n=1$. For many parameters, the litter was appropriately used as the N. For others, the embryo was the N but now with information making it clear that it was from $N=3$ dams, the reader can interpret the data how they see fit. Thank you for providing additional information about the model.

Thanks for your review as your comments have greatly improved the clarity of this manuscript.

Reviewer #3 (Remarks to the Author):

All points from the initial review have been successfully addressed.